# In Vitro Transcribed RNA-Based Platform Vaccines: Past, Present, and Future

**DOI:** 10.3390/vaccines11101600

**Published:** 2023-10-16

**Authors:** Alexey D. Perenkov, Alena D. Sergeeva, Maria V. Vedunova, Dmitri V. Krysko

**Affiliations:** 1Institute of Biology and Biomedicine, National Research Lobachevsky State University of Nizhny Novgorod, 603022 Nizhny Novgorod, Russia; 2Cell Death Investigation and Therapy (CDIT) Laboratory, Anatomy and Embryology Unit, Department of Human Structure and Repair, Faculty of Medicine and Health Science, Ghent University, 9000 Ghent, Belgium; 3Cancer Research Institute Ghent, 9000 Ghent, Belgium

**Keywords:** mRNA vaccine, self-amplifying mRNA vaccine, trans-amplifying mRNA vaccine, circular RNA vaccine, immunogenicity, vaccine

## Abstract

mRNA was discovered in 1961, but it was not used as a vaccine until after three decades. Recently, the development of mRNA vaccine technology gained great impetus from the pursuit of vaccines against COVID-19. To improve the properties of RNA vaccines, and primarily their circulation time, self-amplifying mRNA and trans-amplifying mRNA were developed. A separate branch of mRNA technology is circular RNA vaccines, which were developed with the discovery of the possibility of translation on their protein matrix. Circular RNA has several advantages over mRNA vaccines and is considered a fairly promising platform, as is trans-amplifying mRNA. This review presents an overview of the mRNA platform and a critical discussion of the more modern self-amplifying mRNA, trans-amplifying mRNA, and circular RNA platforms created on its basis. Finally, the main features, advantages, and disadvantages of each of the presented mRNA platforms are discussed. This discussion will facilitate the decision-making process in selecting the most appropriate platform for creating RNA vaccines against cancer or viral diseases.

## 1. Introduction

Vaccines in current use can be broadly categorized into two groups: traditional vaccines, such as live, attenuated and inactivated vaccines, and more modern vaccines, including peptide, DNA, and mRNA vaccines. Each group possesses its own set of advantages and disadvantages. 

Traditional vaccines provide long-term protection against infection, but they require a longer development process and face challenges in scaling up production. In contrast, modern vaccine approaches do not encounter such drawbacks, and among them, mRNA vaccines exhibit several advantages. One primary advantage is the safety of mRNA vaccines. Unlike DNA vaccines, mRNA does not integrate into the vaccinated individual’s genome. Another major advantage is that, unlike attenuated vaccines, mRNA does not pose a risk of infection. On top of that, mRNA vaccines are relatively inexpensive to produce and their production is faster than that of attenuated or inactivated vaccines [1,2]. Moreover, the expression of the protein encoded by mRNA vaccines is relatively short-lived due to natural cellular degradation processes, rendering it easily controlled. Furthermore, unlike DNA vaccines, mRNA vaccines are translated in the cytoplasm without the need to overcome the nuclear barrier [2,3]. Moreover, the problem of the low immunogenicity of peptide vaccines is not an issue in mRNA vaccines [3,4,5]. On top of that, mRNA vaccines can be produced more economically and rapidly than attenuated and inactivated vaccines [1,2]. Finally, the inherent versatility of the key elements in mRNA vaccine composition makes it easy to modify the encoded protein sequence with minimal resources and time investment, thus enabling the development of vaccines with new properties [6].

The main challenges associated with mRNA vaccines are related to the immunogenicity of the RNA molecule itself, which can limit the amount of protein translated in the body. If the immune response is initiated before the mRNA is translated, the vaccine is rendered ineffective. This has been addressed by reducing the administered dose and replacing classical nucleotides with modified ones. The intracellular stability of mRNA vaccines has been augmented by incorporating modified caps, nucleotides, and poly(A) tails, along with optimal regulatory elements. Additionally, the delivery of mRNA vaccines has been enhanced by introducing ionizable lipids [3,7,8,9]. As a result, mRNA vaccines are a promising and innovative direction in the development of preventive and therapeutic vaccines.

In addition to traditional mRNA vaccines, several novel modifications have emerged, including self-amplifying (SAM) mRNA, trans-amplifying mRNA (taRNA), and circular RNA (circRNA). Each of these mRNA-based vaccine varieties comes with its own set of advantages and disadvantages, which adds complexity to the selection of appropriate technology for specific applications. There are already many mRNA-based vaccines against various diseases, including a vaccine against SARS-CoV-2 [3]. Therefore, this review aims to evaluate the key distinguishing features of each vaccine variant (i.e., mRNA, SAM, taRNA, circRNA) to facilitate selection of the most appropriate platform for creating an RNA vaccine against tumors or viral diseases.

## 2. Historical Background

mRNA, as a ribonucleotide template for protein synthesis transcribed from DNA, was initially discovered in 1961 (Figure 1). However, it took several decades of scientific discoveries and experimental investigations before it was used as a vaccine [10]. 

In 1965, A.D. Bangham introduced lipid bilayer particles, later known as liposomes, which played a crucial role in the delivery of mRNA vaccines [11]. Liposomes were first employed for drug delivery in 1971, and only three years later, they were already being used for delivery of diphtheria toxoid in murine models. Concurrently, in 1969, the first laboratory-produced protein was obtained using isolated mRNA. These advances collectively enabled the delivery of mRNA to mouse cells and the subsequent production of the encoded protein using liposomes as early as 1978. 

CircRNA, discovered in 1976, gained popularity as a vaccine several decades later [12]. Eight years after its discovery, a team led by Douglas Melton, under the leadership of Paul Krieg, successfully produced synthetic mRNA using RNA synthetase. In 1988, the first small-molecule synthetic circRNA was obtained by a chemical procedure [13]. 

By 1993, the considerable expertise and knowledge that had accumulated enabled the development of the first mRNA flu vaccine, which was tested in mice. Two years later, a cancer vaccine was also produced using mRNA technology [14]. However, it was not until 2015 that mRNA-based influenza vaccines progressed to clinical trials. This slow progress can be attributed to several limitations of mRNA, such as its instability, short half-life within cells, and relatively high production costs. Consequently, considerable investments were directed toward alternative vaccine technologies considered more promising at that time, which led to the advancement of DNA and protein vaccines [6]. Nonetheless, in 1994, the first SAM mRNA was successfully developed, although its widespread adoption occurred much later [15]. 

It is notable that mRNA vaccines held greater promise for the treatment of oncological conditions compared to viral diseases. Towards the end of the 20th century, Eli Gilboa and colleagues initiated human testing of mRNA cancer vaccines. However, the vaccine candidate for late-stage cancer did not yield the expected outcomes and was not approved, thereby impeding the progress of mRNA vaccine technology. Nonetheless, E. Gilboa’s research served as a catalyst for CureVac and BioNTech to embark on dedicated development of mRNA vaccines in the early 2000s. Studies demonstrated that direct administration of mRNA vaccines triggers a robust immune response against the mRNA, which presented a serious hurdle for their utilization. It was not until 2005 that Kariko and Weissman successfully addressed this challenge by substituting uridine in the vaccine mRNA with pseudouridine, substantially reducing the immunogenicity of the mRNA [14]. In 2011, a significant advancement in mRNA technology known as SAM was achieved with the creation of the first trans-amplifying mRNA (taRNA) [16].

The delivery of RNA vaccines posed another challenge in their development. Initially, cationic lipids with a positively charged head group were utilized; the head group interacts with the negatively charged RNA [17]. However, the sustained positive charge of these lipids was toxic within the body, preventing their use in vaccines [18]. In 2012, Pieter Cullis and his team started experimenting with various lipid compositions of liposomes for RNA delivery. The funding provided by the US Defense Advanced Research Projects Agency for the study and development of RNA vaccines in the same year added impetus to the progress of mRNA vaccine technology [14]. Ultimately, Cullis and his team successfully developed an optimal lipid formulation that incorporated ionizable lipids capable of transitioning from a positive to a neutral charge under physiological conditions, reducing the toxicity of liposomes significantly, while extending the storage duration of mRNA vaccines within this lipid-based delivery system [18].

These important discoveries culminated in the groundbreaking work of Ugur Sahin in 2017, who developed a personalized mRNA vaccine against melanoma. Sahin’s research yielded promising results in therapy and ignited considerable interest in the advancement of mRNA-based cancer vaccines [19]. While mRNA-based therapeutic vaccines for cancer demonstrated notable progress, prophylactic vaccines against viral infections using mRNA technology lagged behind. Notably, Moderna, one of the leading mRNA vaccine companies, had developed nine mRNA vaccines for various human infectious diseases by 2020, but none achieved significant success. Other companies involved in the development of mRNA-based vaccines had similar experiences. However, the emergence of the SARS-CoV-2 virus in 2019 dramatically altered the landscape, motivating large companies to leverage their accumulated experience and swiftly create mRNA-based vaccines in response to the global pandemic [14,20]. As early as one year after the onset of the pandemic, promising mRNA prototypes of vaccines against SARS-CoV-2 virus were generated [3]. As a result, mRNA vaccines are currently undergoing rapid development and closing the technological gap with DNA vaccines, which were developed much earlier.

## 3. Non-Amplifying In Vitro-Transcribed mRNA Platform

Before mRNA synthesis in vitro, the DNA template needs to be linearized. This is typically done with restriction enzymes or by performing PCR to obtain the desired DNA fragment in a linear form suitable for transcription. In vitro-transcribed mRNA (IVT mRNA) is synthesized with bacteriophage T3, T7, or SP6 RNA polymerase [21]. A specific sequence recognized by the corresponding polymerase is incorporated at the 5’ end of the DNA template. For instance, if bacteriophage T7 is to be used, the sequence 5’-TAATACGACTCACTATA-3’ is commonly employed [22] (Figure 2).

A mRNA molecule comprises five essential elements: 5′ cap, 5′-untranslated region (5′-UTR), open reading frame (ORF), 3′-UTR, and 3′ poly(A) tail (Figure 3). Each of these elements plays a crucial role in its function as a template for protein synthesis [21,23].

### 3.1. 5′ Cap Structure

The 5′ cap is a methylated guanoside structure that serves multiple important functions in mRNA processing and stability (Figure 3). It facilitates pre-mRNA splicing, export of mRNA from the nucleus to the cytoplasm, translation initiation, and protection against exonucleases. In eukaryotes, the simplest form of the cap structure is 7-methylguanosine (m7G), which is linked to the mRNA molecule by a 5’-5’-triphosphate bridge. This minimal structure at the 5’ end of mRNA is referred to as Cap 0 (m7GpppNp, where p represents a phosphate group and N denotes any mRNA nucleotide). Cap 0 formation occurs during mRNA transcription in the nucleus [24]. Cap 0 is recognized by the eukaryotic translation initiation factor 4E (eIF4E) and is commonly found in plants (including plant viruses) and fungi. In animals, however, Cap 0 is relatively rare and is found only in certain organisms, such as *Drosophila melanogaster*. Consequently, mammalian cells may perceive Cap 0 mRNA as foreign and trigger an inflammatory response, which might impede the translation of the vaccine RNA [25]. Cap 0 undergoes modification by Cap 2’-O-methyltransferase 1 (CMTR1) in the cell nucleus, resulting in the formation of m7GpppNm2’-Op, known as Cap 1. Cap 2 (m7GpppNm2’-OpNm2’-Op) differs from Cap 0 by the additional CMTR2-catalyzed methylation of the first and second mRNA nucleotides at the 2’-O-position of the ribose [26]. In animals, Cap 1 and Cap 2 are typical, and the ratio between them in a cell depends on the species. In mammals, certain cells may also contain N6-methyladenosine (m6A), and the first transcribed nucleotide can be m6Am instead of m7G. The level of protein production using mRNAs containing different types of caps largely depends on the specific cellular context into which the RNA is introduced. Studies have demonstrated that mRNA with Cap 1 generally yields higher protein levels compared to Cap 0, although no distinct advantages have been established in comparison to Cap 2. Therefore, the choice of cap modification depends on the particular cell line or cellular environment in which the mRNA is to be delivered [27,28]. Another important function of the cap is to protect mRNA from degradation by 5’-end RNases, such as the XRN1 family [29].

In the past, mRNA was capped by sequential enzymatic reactions catalyzed by RNA 5’-triphosphatase (RTPase), guanylyltransferase (GTase), and N7-methyltransferase (N7MTase). This set of enzymes results in the formation of IVT (in vitro-translated) mRNA containing Cap 0 [26]. However, more recently, Vaccinia Capping Enzyme (VCE) has become the preferred method because it combines the activities of all three enzymes (RTPase, GTase, N7MTase), resulting in the direct formation of a Cap 0 in a single step (Table 1). To produce Cap 1 mRNA, an additional enzymatic reaction with mRNA cap 2’-O-methyltransferase (MTase) is required. The efficiency of this capping process can reach 100% [30]. Consequently, the creation of Cap 1 mRNA entails the synthesis of the mRNA itself, as well as two supplementary enzymatic reactions, thereby prolonging the procedure and increasing its cost. To address this issue, synthetic analogs of the cap have been developed to enable co-transcriptional capping. The first variant of a cap analog was m7GpppG, which could successfully incorporate into the 5’ end of IVT mRNA [31]. However, a significant drawback of the cap m7GpppG analog is its ability to attach in the reverse orientation, leading to the formation of non-functional RNAs [32]. One of the early successful alternatives was the Anti-Reverse Cap Analog (ARCA), which is a 3’-O-Me-m7G(5’)ppp(5’)G and serves as an analogue of Cap 0 with the correct orientation, since the 3’-OH group is methylated and does not allow attachment to the 3’ end of the mRNA [33]. Studies have indicated that the protein production level in mRNA containing the conventional Cap 0 was slightly higher than that of mRNA containing ARCA. To obtain Cap 1, an additional reaction is performed by adding 2’-O-methyltransferase to ARCA mRNA. Subsequently, an analogue of the cap (CleanCap^®^) was introduced by TriLink BioTechnologies (San Diego, CA, USA). It incorporates co-transcriptionally and represents Cap 1 [34]. Numerous analogs of CleanCap^®^ have been developed, such as CleanCap^®^ Reagent AG 3’OMe (m7G3’OMepppA2’OMepG), CleanCap^®^ Reagent AG (m7GpppA2’OMepG), and CleanCap^®^ Reagent AU (m7GpppA2’OMepU) [35]. Studies have demonstrated that the use of CleanCap^®^ cap analog and its modifications leads to a significant increase in the proportion of mRNAs containing cap structures (up to 99%) compared to ARCA (approximately 70%) [36]. The manufacturer of CleanCap^®^ claims that these mRNAs are less immunogenic and produce more protein compared to Cap 0 [37]. However, recent investigations have indicated that the increased protein production observed in mammalian cells with high doses of the CleanCap^®^ AG cap analog is accompanied by the heightened production of pro-inflammatory cytokines when compared to ARCA and also involves higher costs [38]. Ongoing research is currently focused on exploring and evaluating new analogs of the cap, with promising results [29].

### 3.2. 5′-Untranslated Region Structure

The 5′-UTR of mRNA plays a crucial role in regulating protein translation (Figure 3). The significance of the 5’-UTR is highlighted by the rates of ribosome association with mRNA, which can differ by orders of magnitude, depending on the sequence [39]. In eukaryotes, the average length of the 5’-UTR ranges from 53 to 218 nucleotides, with a median of 210 nucleotides in humans [40]. The 5’-UTR serves as a platform for transcriptosome assembly, and its sequence can either enhance or hinder translation. Eukaryotic translation initiation begins with the assembly of the 43S pre-initiation complex (PIC), which includes methionyl initiation tRNA (Met-tRNAi) in complex with eukaryotic initiation factor 2 (eIF2). eIFs 1, 1A, 3, and 5 bind to the PIC complex, followed by mRNA interaction facilitated by the eIF4F complex recognizing the cap structure. The eIF4F complex consists of eIF4A, eIF4G, eIF4E RNA helicases, and poly(A)-binding protein (PABP). The PIC scans the mRNA in the 5’-3’ direction until it encounters an AUG start codon that is complementary to Met-tRNAi. The interaction between Met-tRNAi and AUG triggers the binding of the 60S ribosomal subunit, forming a complete 80S ribosome ready for protein synthesis. The context of the AUG codon’s location is a critical determinant of the interaction between Met-tRNAi and the start codon. If the first codon is situated in an “unfavorable” nucleotide environment, Met-tRNAi may “slip” past it. In mammals, the optimal context for the AUG codon is known as the Kozak consensus sequence, which consists of 5’-GCCRCCATGG-3’ [41,42].

Secondary structures of mRNA play a significant role in the regulation of translation. The presence of a hairpin structure following the AUG codon can impede the movement of the PIC complex, thereby reducing the likelihood of “slippage” even in unfavorable nucleotide contexts. Conversely, if a hairpin structure is located within the coding region of the mRNA sequence, it typically leads to a decrease in translation efficiency. Mammalian 5’-UTRs, in particular, contain numerous secondary structures, making them highly dependent on the activity of the eIF4A RNA helicase, which unwinds these structures [43].

In addition to the conventional translation initiation mechanism, there is an alternative mechanism by which the PIC interacts with internal ribosome entry sites (IRES). These IRES elements are found in many viral mRNAs, enabling them to recruit the 40S ribosomal subunit and PIC to internal sites within the 5’-UTR, bypassing the cap structure. Interestingly, IRES elements are also present in various mRNAs involved in the cellular stress response in mammals [44]. 

In developing mRNA vaccines, it is crucial to avoid secondary structures in the 5’UTR region, particularly those rich in GC content, and to exclude AUG codons to prevent premature translation initiation [45]. An elevated adenine content in the 5’UTR can trigger cap-independent translation initiation but may result in rapid mRNA degradation in the absence of translation [46].

Among the popular choices for 5’UTRs are those derived from the human or Xenopus laevis α- and β-globin genes, which produce good protein expression levels [47]. A further advancement in the design of 5’UTR involves the exploration of synthetic sequences. Through bioinformatics, 280,000 randomized 5’UTRs have been generated and analyzed, with 10 sequences showing promising potential [48]. Another approach is the utilization of minimal 5’UTRs consisting of 12–14 nucleotides, which has also shown promising outcomes [49].

Optimal 5’UTRs are being sought for incorporation into mRNA vaccines, including the exploration of highly successful viral IRES elements as potential candidates. One of the challenges encountered when using ARCA for creating 5’UTR in IVT mRNAs is the relatively low efficiency of capping. Moreover, the manufacture of Vaccinia Capping Enzyme (VCE) and m7GmAmG enzymes is expensive. Consequently, viral internal ribosome entry sites (IRES) present a viable alternative for initiating cap-independent translation [50]. Four classes of viral IRES have been identified, with each class recruiting different translation initiation factors (Figure 4) [51]. Class I IRES utilize all initiation factors except for eIF4E, which is typically involved in cap recognition. Notably, picornaviruses, including Coxsackievirus B3 (CVB3) and some rhinoviruses such as human rhinovirus (HRV), demonstrate particularly robust Class I IRES activity in IVT mRNA [50]. In contrast, Class II IRESs directly recruit translation factors to the AUG codon, bypassing the need for mRNA scanning by the pre-initiation complex, which distinguishes it from Class I IRES. Class II IRESs are exemplified by viruses such as encephalomyocarditis virus (EMCV) and foot-and-mouth disease virus (FMDV). On the other hand, Class III IRESs share similarities with Class II IRESs but recruit translation initiation factors specifically at the AUG codon by a mechanism that primarily involves the pre-initiation complex (PIC) and eIF3. Hepatitis C virus (HCV) and Japanese encephalitis virus (JEV) are representative examples of Class III IRESs. Class IV IRESs, on the other hand, exhibit a remarkable independence from eIFs and directly recruit the 40S subunit, thereby facilitating translation initiation. These IRESs structurally mimic Met-tRNAi. Cricket paralysis virus (CrPV) and Taura syndrome virus (TSV) are prominent examples of Class IV IRESs [50,51,52,53]. It is noteworthy that viral IRESs incorporated into IVT mRNAs have demonstrated higher protein production levels compared to cellular IRESs such as eukaryotic initiation factor 4G (eIF4G), fragile X Messenger Ribonucleoprotein 1 (FMR1), or connexin 43 (CN43) [50]. However, a limitation of viral IRESs is that they are longer than the typical cap-dependent 5’UTRs, which usually do not exceed 50 nucleotides.

### 3.3. Open Reading Frame Structure

The ORF in mRNA is the encoded sequence that determines the functional properties of a vaccine (Figure 3). In many cases, the first region of the ORF encodes a signal peptide, typically up to 30 amino acids long. The signal peptide plays a crucial role in determining the localization of the synthesized protein. For therapeutic vaccines, the desired localization may be the cell membrane or secretion as exosomes. One commonly employed approach in therapeutic vaccines is the use of MHC class I signal peptide fragments. For optimal results, an N-terminal signal peptide (SP) is combined with an MHC class I trafficking signal (MITD) fused to the C-terminus of the antigen. This combination significantly improves the antigen presentation of the epitopes of interest in MHC class I and class II in human and mouse dendritic cells [54].

In some cases, tissue plasminogen activator (tPA) is used along with MITD at the N-terminus of the target antigen. tPA may also be used without MITD [3]. The inclusion of additional signals aims to stimulate the CD4+ T-lymphocyte response. CD4+ T-lymphocytes are activated upon recognition of antigens presented by MHC class II, which is primarily responsible for presenting exogenous peptides. 

The expression of IVT mRNA suggests an endogenous arrangement of proteins, which primarily stimulates the presentation of peptides in MHC class I, and, accordingly, CD8+ T-lymphocytes [55]. Early investigations have demonstrated that the incorporation of MHC class I signal peptides into vaccine proteins facilitates their intracellular trafficking to key compartments associated with MHC class II processing, including the endoplasmic reticulum, Golgi apparatus, endosomes, lysosomes, and the cell plasma membrane [54]. Consequently, these signal sequences contribute to the robust development of CD4+ T-lymphocyte immune responses elicited by the vaccine (Figure 5). 

Another notable advantage of localizing vaccine proteins on the cell membrane surface is the concurrent stimulation of B-lymphocytes. B-lymphocytes typically scan surface-displayed native antigens on antigen-presenting cells via their receptors, thus enabling their activation and the subsequent initiation of a B-cell response [56].

Adjuvants play a crucial role in IVT RNA vaccines. One example is the use of the CD40 molecule to effectively stimulate antigen-presenting cells such as dendritic cells and macrophages, as well as CD8+ T lymphocytes and B lymphocytes. The use of CD40 as an adjuvant demonstrates its ability to stimulate all key components of the immune system necessary for the establishment of robust protection following vaccination [57]. Another strategy involves the incorporation of separate IVT mRNAs encoding adjuvants. A mix of three mRNA molecules (TriMix) vaccines have shown promising results in this regard. TriMix vaccines comprise mRNAs from three distinct adjuvants: CD40L, CD70, and TLR4, or IL-23, IL-36, and OX40L [58,59].

The selection of codons used in the coding region of a vaccine is an important consideration, particularly for allogeneic nucleotide sequences that might contain codons rarely used in human cells, such as those found in viral RNA. To enhance the expression of such proteins in human cells, synonymous mutations are introduced, artificially modifying certain nucleotides. These modified sequences, known as humanized sequences, are then used for protein expression [60,61]. In mammals, the third nucleotide in a triplet is usually cytosine or guanine. The presence of uracil or adenine in this position usually reduces the efficiency of protein expression, and studies have demonstrated that replacing uracil in the third position with cytosine or guanine can increase protein expression up to fivefold [56]. However, it is important to note that accelerating protein synthesis can sometimes disrupt proper folding. On the other hand, natural RNAs may contain rare codons, which may slow down the rate of protein synthesis but enable correct folding [62]. Therefore, when optimizing codons, close attention should be paid to RNA sequences encoding proteins that rely on complex folding processes, while IVT RNAs encoding proteins that do not require intricate folding can be optimized more freely. 

Immunological bioinformatics is currently employed to determine the primary sequence of a vaccine, which plays a crucial role in its immunogenic properties [63]. This computational approach can predict most immunogenic epitopes, making it possible to select peptides as vaccine candidates rather than entire proteins.

The immunogenicity of epitopes is evaluated on three lymphocyte populations involved in long-term vaccine epitope memory: cytotoxic T lymphocytes (CTL), helper T lymphocytes (HTL), and linear B lymphocytes (LBL) [3,64]. Epitope selection is tailored for each cell population due to significant variations in their recognized fragment sizes. Consequently, the in silico approach significantly reduces vaccine development time and minimizes the size of the sequence encoded in IVT mRNA [65]. 

In addition, multi-epitope vaccines incorporate 10 or more immunogenic antigen fragments, each consisting of approximately 30 amino acids. This approach significantly reduces the cost of vaccine production without compromising its effectiveness [66]. Multi-epitope vaccines enable the inclusion of multiple distinct antigens within a single vaccine [67]. In a multi-epitope vaccine, the epitopes are separated by a linker or spacer sequence of low immunogenicity [66]. Linkers can be categorized into three groups: in vivo cleavable, rigid, and flexible [68]. Commonly used linkers include EAAAK, GPGPG, GGGGS, KK, and AAY. The EAAAK linker is often used to connect epitopes to adjuvants, AAY or GGGGS for separating CTL epitopes, GPGPG for separating Th epitopes, and KK or EAAAK for separating LBL epitopes [3,66,69,70,71,72].

### 3.4. 3′-Untranslated Region Structure

The 3′-UTR is an untranslated region of mRNA located at its 3’ end (Figure 3). The 3’-UTR plays a crucial role in mRNA localization, stability and translation efficiency, making it important in the design of IVT mRNA platforms [73]. The length of the 3’-UTR ranges from a few to hundreds of nucleotides in different organisms [74,75]. Notably, the length of the 3’-UTR can affect the level of protein expression, with longer fragments exhibiting shorter half-lives and shorter fragments generally having lower translation efficiency [76]. The 3’-UTR used in the creation of IVT mRNA is typically 130‒280 nucleotides long. Initially, 3’-UTRs derived from human hemoglobin subunits α (HBA) and β (HBB) mRNAs were commonly used. 

However, more recently, it has been discovered that incorporating two repeats of the 3’-UTR of human β-globin (2hBg) in mRNA improves its stability and increases protein production [77]. 2hBg has become the gold standard for 3’-UTR constructs for in vitro mRNA synthesis. Subsequent research on 3’-UTR gene fragments has identified other effective options. For example, the mitochondrially encoded 12S rRNA (mtRNR1) has shown high protein levels when used alone or in combination with an amino-terminal enhancer of split (AES) gene fragment [76]. Various IVT mRNA manufacturers have successfully utilized 3’-UTR fragments from genes such as hydroxysteroid dehydrogenase (3β-HSD), tyrosine hydroxylase (TH), heat-shock protein 70 (Hsp70), dynein heavy chain 2 (DNAH2), and many others [78]. The future development of 3’-UTR involves the creation of synthetic fragments that may surpass the expression levels of existing natural analogs. However, the progress in creating artificial sequences is hindered by the limited research on the 3’-UTR compared to the 5’-UTR (5’ untranslated region) due to methodological limitations [73]. It has been observed that AU-rich regions in the 3’-UTR can significantly shorten mRNA lifetime and should be avoided when designing IVT mRNA [73,78]. However, AU-rich regions also contribute to the rapid induction of protein expression when immune cells are stimulated with lipopolysaccharide [79].

### 3.5. Poly(A) Tail Structure

The poly(A) tail, the 3’ end of the mRNA, is characterized by multiple adenine repeats (Figure 3) added to the mRNA molecule after transcription. In the cytoplasm, the poly(A) tail is recognized by the poly(A) binding protein (PABP), which requires at least 12 adenine residues on the poly(A) tail. PABP interacts with the cap structure of mRNA, facilitated by eIF4G, to form a translation complex [30,80]. Another function of the poly(A) tail is to protect the mRNA from degradation by exonucleases, thereby increasing its stability and prolonging its lifespan. This contributes to higher levels of protein production [80]. The length of the poly(A) tail typically ranges from 100 to 300 nucleotides and varies with the cell type. For instance, in dendritic cells, the poly(A) tail is 120–150 nucleotides long, while in T-lymphocytes, it is generally about 300 nucleotides [81,82]. Translation efficiency drops significantly when the poly(A) tail is shorter than 20 nucleotides [83,84].

The poly(A) tail can be generated in vitro using *E. coli* Poly(A) Polymerase I (E-PAP) or a DNA template. Enzymatic polyadenylation with E-PAP occurs as a separate step after transcription and can produce a poly(A) tail longer than 100 nucleotides. However, one significant drawback of using polyadenylation enzymes is the unpredictability of the poly(A) tail length, which prevents standardization and necessitates a separate adenylation step after transcription [85]. Another challenge is the increased frequency of hydrolysis of IVT mRNA during enzymatic polyadenylation, which occurs at pH > 7.5. IVT mRNAs longer than 3000 nucleotides are particularly sensitive to alkaline hydrolysis [86]. 

A poly(A) tail can also be added to mRNA in vitro by PCR or by using a plasmid DNA template. In the PCR-based method, a poly(A) tail is generated by incorporating a reverse poly(T) primer of a specific length during the PCR reaction. This method enables precise control of the length of the poly(A) tail, but it is not suitable for producing more than a few hundred milligrams of polyadenylated RNA. Additionally, there is a higher risk of mutagenesis during template amplification. Therefore, this method is commonly used in laboratory research but is not suitable for large-scale production. Using a plasmid DNA template is more scalable and cost-effective for in vitro transcription of up to grams of mRNA. This method also carries a lower risk of mutagenesis compared to PCR. However, one challenge associated with using plasmid DNA templates is the potential for recombination within the homopolymeric regions during plasmid amplification. Deletions in the plasmid sequence of IVT mRNA can occur, particularly when the poly(A) tail exceeds 100 nucleotides. At lengths greater than 150 nucleotides, it becomes difficult to obtain bacterial clones containing the complete, intended poly(A) tail sequence [52,82,87]. One approach to preserve a long poly(A) tail while ensuring high translation levels and avoiding deletions in bacterial cells is to introduce a linker within the tail sequence. This linker is typically one to ten nucleotides long and is not composed of adenine, effectively dividing the poly(A) tail into two approximately equal parts of about 60 nucleotides each [82,88,89]. An example of this approach can be seen in mRNA vaccines developed by Pfizer/BioNTech, which incorporate a spacer in the middle of the poly(A) tail [90,91]. The optimal length of a long poly(A) tail in IVT mRNA is 120–150 nucleotides, which is characteristic of vaccines targeting dendritic cells. This assumption is supported by experiments conducted on dendritic cells, where the elongation of the poly(A) tail in IVT mRNA is directly correlated with increased protein levels in cells persisting for up to 150 h after transfection [47]. On the other hand, mRNAs that are translated at a high rate typically have shorter poly(A) tails of 30–60 nucleotides and have longer half-lives [92]. Studies have demonstrated that mRNAs transcribed in vitro with synthetic 5’UTRs designed for high ribosome loading, despite having a poly(A) tail as short as 40 nucleotides expressed higher relative protein levels than mRNAs with the classic β-globin 5’UTR, which has a poly(A) tail of 148 nucleotides [47]. These findings challenge the conventional paradigm regarding the length of the poly(A) tail of IVT mRNA vaccines. Another paradigm that can also be reconsidered is the nucleotide composition of the poly(A) tail. The tail is known to contain only adenines, but studies show that the inclusion of other nucleotides increases the efficiency of mRNA translation [93].

## 4. Self-Amplifying IVT mRNA Platform

Two types of IVT mRNAs are utilized in the development of therapeutic and prophylactic vaccines: non-amplifying mRNA (NRM) and self-amplifying mRNA (SAM) (Figure 6) [94]. 

The concept of self-amplifying mRNA was first applied in 1994 using Semliki Forest virus (SFV) by Zhou et al. [15]. Other well-studied alphaviruses such as Sindbis virus and Venezuelan equine encephalitis virus are also employed for SAM production [95]. SAM, like NRM, consists of a 5’ cap, 5’-UTR, long open reading frame (ORF) region, 3’-UTR, and 3’ poly(A) tail. However, SAM differs significantly from other vaccine types because it can self-replicate by means of the replicase located downstream of the 5’ UTR. The replicase includes four non-structural proteins (nsP1, nsP2, nsP3, and nsP4) derived from alphaviruses [96]. nsP1 possesses GTase and N7MTase activities, while nsP2 exhibits RTPase activity, enabling the capping of IVT mRNA to form Cap 0 [97]. nsP2 also functions as a protease and helicase, facilitating the processing of the entire nsP complex. The role of nsP3 is not fully understood, but it interacts with various host cell proteins, contributing to the suppression of the antiviral response [97,98]. The most conserved protein among alphaviruses is nsP4, which functions as an RNA-dependent RNA polymerase (RdRp) responsible for amplifying the number of copies of the original IVT mRNA [99]. After the SAM enters the cytoplasm of the target cell, the protein translation process is initiated, primarily involving the nsPs, which are located upstream of the main protein coding region. The nsPs form an early replication complex that utilizes the SAM template to generate a complementary RNA strand. Subsequently, the nsP polypeptide is cleaved into individual proteins, and a late replication complex is formed. This late replication complex is responsible for synthesizing a copy of SAM on the RNA template, thereby increasing the initial number of IVT RNA copies. One notable distinction between SAM and NRM is the presence in SAM of an alphavirus subgenomic promoter (SGP) region located before the gene of interest (GOI) in the main protein coding sequence. The SGP facilitates the initiation of GOI transcription by bypassing the reading of the sequence encoding viral nsP proteins. This mechanism promotes the formation of mRNA copies containing only the GOI, which are then translated [78,98,99,100,101,102]. This replication mechanism allows for a 30 to 1000-fold reduction (0.1–10 μg versus 30–100 μg) in the required dose of SAM compared to NRM drugs [103]. Consequently, the lower dose requirement of SAM is less immunogenic, safer, and has a longer duration of action, which contributes to the development of a robust immune response [104], and also requires less raw material to synthesize. Indeed, SAMs do have some of the drawbacks associated with the inclusion of viral sequences. The length of the nsP fragment, which is approximately 7 kb, imposes limitations on the length of the gene of interest (GOI) and complicates the creation of a vector for bacterial propagation and vaccine packaging. Additionally, viral nsPs can have an impact on the host cell and trigger immune hyperactivation [78,100,104]. Another characteristic of SAM is the generation of intracellular double-stranded RNA (dsRNA) during replication, which activates innate immunity and can interfere with protein translation [105]. Efforts have been invested in optimizing the codons of the viral replicase to enhance its translation rate [106,107], thereby improving the properties of SAM. 

Several vaccines based on SAM are in pre-clinical trials. These vaccines target pathogens such as SARS-CoV-2, influenza virus, rabies virus, Zika virus, Ebola virus, VEEV (Venezuelan equine encephalitis virus), HIV-1, and certain microbes causing bacterial infections or parasitic infestations, as well as cancer [108,109,110,111,112,113,114,115,116,117]. 

## 5. Trans-Amplifying IVT mRNA Platform

taRNA is a form of SAM wherein the viral sequences, nsPs, and GOI are contained in different mRNAs but are used together (Figure 6) [106]. The viral replicase can be either nrRNA or saRNA, and the mRNA encoding the GOI is called trans-replicon (TR)-RNA. To achieve TR-RNA amplification, conserved sequence elements (5’CSE and 3’CSE) derived from the alphavirus flank the GOI, with SGP of the alphavirus upstream of the GOI [107]. The concept of the trans-replication system was initially proposed by Pirjo Spuul et al. in 2011 [16]. This design encompasses the advantages of SAM while mitigating certain drawbacks. Notably, by encoding the replicase separately in an RNA platform, the limitation on the GOI length is circumvented and the use of modified nucleotides is not constrained. Additionally, this approach is safer because it reduces the likelihood of producing recombinant virus particles [118]. Intriguingly, taRNA is more immunogenic, so lower doses than saRNA (as low as 50 ng) are sufficient to elicit a comparable immune response [106]. Recently, a variation of taRNA has been reported. It includes two additional TR-RNAs alongside the replicase-encoding RNA [107]. Ongoing progress in taRNA technology has led to the proposal of a modified version that incorporates adenine-rich regions in the 5’UTR. Notably, this modified taRNA lacks the alphavirus SGP, resulting in a shorter RNA and a tenfold reduction in vaccine dose without compromising in vitro expression levels [119]. However, a significant drawback of taRNA is the requirement for at least two distinct RNAs, one for the replicase and the other for the GOI [120]. Overall, taRNA technology is still in its early stages but it has promising prospects for practical applications. Pre-clinical investigations are currently underway for a taRNA vaccine targeting the influenza virus [78,106]. Furthermore, a bivalent vaccine targeting Chikungunya and Ross River Virus has been developed using this approach [121].

## 6. Circular RNA Platform

After circular RNA (circRNA) was discovered in cells in 1979 by electronic microscopy, it was initially regarded as a by-product of erroneous splicing [122,123]. But its vaccine potential was recognized in 2015 with the discovery of translated circRNAs in fruit flies [124], and it has gained recognition as a vaccine platform. 

CircRNA is self-circularized and therefore lacks traditional 5’ and 3’ ends (Figure 6), a conventional cap structure, and a poly(A) tail. This structure makes it resistant to exonucleases, such as RNase R, and enhances its stability. It has a half-life of up to 84 h, in contrast to the average half-life of <20 h for linear mRNA [125,126,127]. Translation is initiated by a mechanism mediated by the IRES [51,78,128,129]. A commonly utilized IRES for translating synthetic circRNA is the encephalomyocarditis virus (EMCV) IRES, which is efficient in various cell lines [51,130,131], as well as those derived from coxsackievirus B3 (CVB3) and human rhinovirus B3 (HRV-B3), along with their modified versions, which have translation levels 3–4 times higher than that of EMCV IRES [131,132,133]. The crucial step in IVT circRNA synthesis is the circularization of linear RNA molecules by chemical or enzymatic methods [134]. The chemical approach for IVT circRNA generation, which originated in 1988, can be regarded as a pivotal milestone in the field. This method uses cyanogen bromide (BrCN) or 1-ethyl-3-(3-dimethylaminopropyl) carbodiimide (EDC) for the formation of bonds between the 5’-terminal phosphate and 3’-terminal hydroxyl groups of the linear RNA. However, this reaction may yield circRNA variants containing 2’-5’ phosphodiester bonds at the junction of the linear molecule, which are considered by-products [127]. The method is no longer commonly employed due to its high cost and low yield, and its applicability to circularization of RNA of only up to 70 nucleotides in length [135]. 

Enzymatic approaches for RNA circularization use bacteriophage T4 enzymes or ribozymes [134]. For the creation of IVT CircRNA, three bacteriophage T4 enzymes are used: T4 DNA ligase (T4 Dnl 1), T4 RNA ligase 1 (T4 Rnl 1), and T4 RNA ligase 2 (T4 Rnl 2) (Figure 7). To circularize the ends of a linear IVT RNA molecule using T4 bacteriophage enzymes, a nucleoside monophosphate must be present at the 5’ end bonded to the OH group at the 3’ end of the RNA [136]. Typically, IVT RNA contains guanosine triphosphate (GTP) at the 5’ end, as nucleoside triphosphates are added during the reaction. There are three strategies to obtain 5’-guanosine monophosphate (GMP) at the 5’ end of IVT RNA:

(1) Addition of an excess amount of guanosine monophosphate (GMP) compared to GTP in the IVT reaction, typically in a ratio of approximately 5/1, increases the likelihood of GMP being present at the 5’ end of the RNA [130,135,137].

(2) Treatment of IVT RNA with *E. coli* 5’-RNA pyrophosphohydrolase (RppH) or similar phosphatases cleaves phosphate groups from the 5’-terminal GTP, resulting in the formation of a 5’-terminal GMP [138,139].

(3) Treatment of IVT RNA with alkaline phosphatases to remove all phosphate groups from GTP, followed by treatment with T4 polynucleotide kinases to phosphorylate guanosine and generate GMP [134,135].

The subsequent step in the generation of IVT circRNA involves the addition of T4 bacteriophage enzymes to the linear RNA with a 5’-terminal GMP. T4 Dnl1, initially employed in 1992, is the first enzyme used for circRNA formation. T4 Dnl1 is used by bacteriophages to repair breaks in double-stranded DNA substrates [140]. A necessary condition for T4 Dnl1 is the presence of a double-stranded nucleic acid substrate. However, since IVT RNA is single-stranded, the reaction requires a short single-stranded DNA molecule complementary to the RNA site being joined. This single-stranded DNA (ssDNA) typically consists of at least 20 nucleotides, 10 of which are complementary to the 3’-end of the IVT RNA and the other 10 complementary to the 5’-end. This positions the ssDNA in the middle of the junction between the ends to form a double-stranded structure [137,141,142]. It should be noted that the efficiency of the RNA ligation reaction is relatively low because T4 Dnl1 is specific for double-stranded DNA (dsDNA) and exhibits weaker binding to RNA/DNA heteroduplexes, necessitating an increase in enzyme concentration in the reaction [135]. 

Due to the low yield and high consumption of T4 Dnl1, this method is now rarely used. Conversely, T4 Rnl 1, which was first used in 1999, is commonly employed as a ligase for circRNA synthesis [134]. T4 Rnl 1 exhibits an affinity for single-stranded RNA molecules (ssRNA) but possesses low specificity [143]. The efficiency of end-joining is primarily influenced by the nucleotide composition of the ligated RNA ends, because adenosine is optimal at the 3’ end and cytidine at the 5’ end [144]. However, IVT RNA typically contains guanosine at the 5’ end, which is the least favorable nucleotide for ligation in this position and thereby complicates the enzymatic reaction. Consequently, IVT circRNA of 6–8 nucleotides is typically produced, as increasing its length significantly reduces the reaction efficiency [145]. The close proximity of the 3’ and 5’ ends that are to be ligated is crucial for the activity of T4 Rnl 1 [146]. However, as the length of IVT circRNA increases, the likelihood of interaction between the RNA ends and the formation of double-stranded RNA (dsRNA) and secondary structures also increases, significantly reducing the efficiency of end ligation with this enzyme [137,147]. To remedy this issue, DNA complementary to IVT RNA is introduced into the reaction, which brings the 3’ and 5’ ends of the RNA in close proximity, with several non-complementary nucleotides at the ends, creating a short single-stranded fragment for ligation [148]. Recently developed is an optimal motif that forms an RNA hairpin structure, effectively bringing the 3’ and 5’ ends closer together. This approach enabled Carmona et al. (2019) to successfully obtain IVT circRNA of up to 4 kb in length [149].

The final enzyme used in the creation of IVT circRNA is T4 Rnl 2, which was first employed in 2002 [134]. Like T4 Rnl 1, this ligase can ligate ssRNA into a circular molecule, but it exhibits greater activity in the ligation of dsRNA molecules [145,150]. To generate dsRNA from a linear IVT RNA molecule, secondary structures are employed at the ligation site, and an RNA or DNA fragment complementary to the ends of the IVT molecule is added. However, the primary challenge remaining in the use of T4 Rnl 2 and T4 Rnl 1 is the low efficiency of the reaction and the increased production of by-products as the length of the coupled IVT RNA increases [134,135]. More recently, a strategy has been proposed to generate IVT circRNAs longer than 1.5 kb. Liang Qu et al. (2022) employed bioinformatic methods to identify a suitable site for RNA ligation within the CVB3 IRES (385 nucleotides). At this site, a favorable secondary structure is formed, enabling high-efficiency ligation with T4 Rnl 2. Using this approach, the authors successfully developed a highly effective vaccine against SARS-CoV-2, which was tested in mice and monkeys [132]. 

Significant progress has been made to resolve the primary challenge associated with the use of T4 bacteriophage enzymes, which is the limited length of IVT circRNA. However, a persistent problem is the formation of RNA concatemers, as most of the ligases used can cross-link ssRNAs into long molecules. To mitigate this problem, the volume of the reaction mixture was increased to reduce the likelihood of close proximity between linear IVT RNA molecules [139,151]. 

Ribozymes are RNAs that possess enzymatic activity, and this unique property is harnessed in the creation of IVT circRNA. Ribozyme sequences facilitate self-splicing, enabling the conversion of linear RNA molecules into circRNA without the need for additional enzymes (Figure 8). The self-splicing process involves two successive transesterification reactions at specific sites, ensuring the production of the desired circRNA product [134]. Three methods are commonly employed for generating circRNA using ribozymes: the group I intron self-splicing method, also known as permuted introns and exons (PIE), the group II intron self-splicing method, and the hairpin ribozyme method [134,135]. The group I intron method is the most widely used in circRNA design [131]. The PIE method was first employed in 1992 to create circRNA using group I introns derived from the Anabaena pre-tRNA gene [152]. Two years later, group I introns from the thymidylate synthase (td) gene of bacteriophage T4 were identified, enabling the generation of circRNA from td exons [153]. These two variants of group I introns are still used to generate circRNA [135,154]. 

It has been observed that the composition of group I introns influences the efficiency of circRNA production. The yield of circRNA is notably higher when using group I introns from the Anabaena pre-tRNA gene compared to those from the td gene [131]. In the PIE method, only GTP and Mg2+ need to be added as co-factors in the reaction [153]. The use of group I introns enables the generation of circRNA longer than 5 kb, which is a significant advantage [131,145]. A distinctive characteristic of group I introns is the requirement for an additional exon sequence derived from the same organism as the group I introns, flanking the GOI. The recognition of specific exon sequences by group I introns allows for precise self-splicing according to these markers. The exchange of exons typically ranges from 70 to 180 nucleotides, while the introns themselves are approximately 300‒400 nucleotides long. After self-splicing, the ends are joined within the exon sequence [134,135]. One significant drawback of using group I introns is that allogeneic exon sequences remain after RNA self-splicing, which separate the GOI and IRES. This separation could affect the expression of the IVT CircRNA. However, efforts are underway to improve the limitations associated with group I introns, and there have been promising results [154,155]. The unique characteristics of the PIE method make it possible to generate circRNA not only in vitro but also in vivo, making it a distinctive approach [134,156].

The self-splicing mechanism of group II introns shares a similar principle with the formation of circRNA, but its advantage is using introns without the need for exons [157]. Group II introns are transposable genetic elements present in organelle genomes and bacteria [158]. A distinguishing structural characteristic of group II introns is the presence of six domains, with the first domain playing a role in determining the specificity of self-splicing through its binding to a specific RNA sequence. This method enables the precise joining of linear RNA sequences without the inclusion of foreign exon fragments during circularization. Recently, there has been a growing interest in the development of vaccines using group II introns, which can be seen as a logical progression from the use of group I introns [159]. However, it is important to note that the self-splicing of group II introns forms 2’, 5’-phosphodiester bonds at the ligation site instead of the natural 3’, 5’-phosphodiester bonds, and the precise mechanism of this process is not fully understood [134,135,160]. Nevertheless, studies have demonstrated that IVT circRNAs containing group II introns are highly stable, weakly immunogenic, have low toxicity, and can produce high levels of protein [159].

Another approach for producing IVT circRNA from a linear RNA molecule is the utilization of hairpin ribozyme (HPR). HPR enables the production of circRNA using T7 RNA polymerase on a single-stranded circular DNA template by a rolling circle mechanism. The resulting linear RNA molecule can undergo self-cleavage and ligation to form circRNA. HPR elements have been identified in the genomes of certain viruses, such as the hepatitis delta virus, and have been employed as a platform for the synthesis of IVT circRNA [134,135,161,162,163]. This method is particularly suitable for producing small circRNA molecules (up to 150 nucleotides) with high yields [134,164]. However, the instability of ligation and self-cleavage mediated by HPR can lead to the generation of reaction by-products, and the resulting circRNA may contain HPR fragments that retain catalytic activity. This renders the current HPR method unsuitable for the development of RNA vaccines [134,144].

In conclusion, among the various ribozyme methods available for generating circRNA, the PIE method has been extensively studied and proven to be useful. However, group II introns hold promise for advancements. Ribozyme methods, in general, offer advantages such as cost-effectiveness, as they do not require additional enzymes, and the reactions can occur co-transcriptionally. These features make ribozyme methods more attractive for circRNA synthesis than enzymatic methods.

## 7. Immunogenicity of RNA Vaccines

The immunogenicity of IVT RNA vaccines is a significant challenge [165]. This immunogenicity arises from the absence of nucleotide modifications that are characteristic of natural mRNA, thereby preventing its effective discrimination by the immune system [85,165]. Hence, the triggered immunological reactions destroy the RNA, nullifying the desired effect. Therefore, the immunogenicity of vaccine RNA should be reduced, while the immunogenicity of the proteins translated from the vaccine RNA should be increased to develop persistent immunity. Among the receptors involved in recognizing IVT RNA are pattern recognition receptors (PRRs), which are widely expressed on various immune cells, particularly antigen-presenting cells [30]. PRRs involved in RNA recognition can be categorized into endosomal PRRs and cytoplasmic PRRs. Endosomal PRRs, primarily Toll-like receptors (TLRs), including TLR-3 for double-stranded RNA (dsRNA) and TLR-7/TLR-8 for single-stranded RNA (ssRNA), are predominantly localized within the endosomes (Table 2). Activation of these endosomal TLRs by RNA triggers intracellular signaling cascades leading to the production of type 1 interferon (IFN-I) and pro-inflammatory cytokines [78,166,167]. IFN-I, acting in an autocrine and a paracrine manner, initiates intracellular signaling pathways that induce the expression of IFN-stimulated genes (ISGs), which exert regulatory functions by inhibiting cellular RNA translation and promoting its accelerated degradation [78].

Among the main pattern recognition receptors (PRRs) in the cytosol are nucleotide oligomerization domain (NOD)-like receptors (NLRs), oligoadenylate synthetase (OAS) receptors, RNA-dependent protein kinase (PKR), and retinoic acid-inducible gene-I-like (RIG-I-like) receptors (RLRs). NOD2, a member of the NLR family, plays a crucial role in recognizing ssRNA. RLRs consist of RIG-I (DExD/H-box helicase 58), which interacts with both ssRNA and dsRNA, as well as melanoma differentiation-associated gene 5 (MDA-5) and LGP2 (DEXH-box helicase 58), which recognize dsRNA [168,169]. Like TLRs and NLRs, RLRs activate the synthesis of IFN-I and pro-inflammatory cytokines in response to RNA recognition [78]. Conversely, PRR interacts with dsRNA and phosphorylates eIF2, leading to the repression of translation. Oligoadenylate synthetase binds to dsRNA and facilitates the activation of RNase L, resulting in RNA degradation [170,171]. Thus, it is essential to create IVT RNA that evades recognition by the immune system [172]. 

To address this challenge, various modified nucleotides have been proposed and incorporated during IVT mRNA synthesis to reduce its immunogenicity [173]. Of the 172 proposed modified bases, only some demonstrated the ability to enhance mRNA expression and stability and reduce immunogenicity [174,175]. These modified nucleotides include N1-methyladenosine (m1A), N6-methyladenosine (m6A), 5-methoxyuridine (mo5U), 2-thiouridine (s2U), pseudouridine (ψ), N1-methylpseudouridine (m1ψ), 5-methylcytidine (m5C), 5-methoxycytidine (5moC), and 5-hydroxymethylcytidine (5hmC) [78,167,176,177,178]. The primary objective of incorporating these modified nucleotides, along with natural nucleotides, is to inhibit the activation of PRRs, such as TLR-3, TLR-7, TLR-8, OAS, RIG-I, and PKR, and thereby enhance the translation of IVT mRNA [78,173,179,180,181,182]. However, it is important to note that the same modified nucleotide can have different effects on the translation processes, depending on the cell type [183]. Therefore, when employing modified nucleotides, it is crucial to consider the specific cell type to which the IVT RNA will be targeted and select the nucleotides accordingly [177].

SAM is highly sensitive to the incorporation of modified bases, as they can disrupt the replicase activity, and therefore, they are usually not used. The lower immunogenicity of SAM vaccines compared to NRM is attributed primarily to the significantly lower doses of SAM employed, rather than to the use of modified nucleotides. Notably, modified nucleotides could be used with optimized taRNAs to reduce immunogenicity and decrease vaccine doses without diminishing the response. However, restrictions still exist in the use of modified nucleotides in TR-RNA synthesis, though the RNA encoding the replicase can be modified [104]. Notably, taRNAs offer an additional advantage over SAM because even lower vaccine doses could be used, further reducing immunogenicity [106,118].

Like mRNA, IVT circRNA incorporates modified nucleotides that reduce immunogenicity while minimally affecting translational efficiency. The incorporation of 5% m6A in circRNA significantly decreases its immunogenicity without impairing translation [133]. m6A is a prevalent RNA modification, occurring in approximately 0.5% of cellular RNA, including circRNA [184,185]. Notably, the YTHDF2 protein, which hinders innate immune activation, binds to m6A within endogenous circRNA in cells [186]. Furthermore, even the presence of a single m6A modification near the start codon can initiate cap-independent translation of circRNA by directly interacting with eIF3 [187]. Studies have demonstrated that the inclusion of multiple copies of the GGACU motif containing m6A enables translation initiation without an IRES, thus permitting the substitution of classical viral sequences with short motifs of approximately 20 nucleotides [188]. Overall, the RR(m6A)CH motif has been identified as particularly favorable for initiating cap-independent translation of circRNA [189]. In contrast, the use of m1ψ in IVT circRNA does not alter its immunogenicity but can negatively affect RNA circularization efficiency and translation levels [190].

It is important to note that the immune response to IVT circRNA has not been sufficiently investigated. However, it is postulated that the primary PRRs involved in recognizing circRNA are RIG-I and PKR. RIG-I activates intracellular signaling pathways, including MAVS, leading to the activation of the transcription factor IRF3 and subsequent synthesis of IFN-I, whereas PKR represses RNA translation [186]. It is hypothesized that the immunogenicity of IVT circRNA primarily arises from residual exons resulting from PIE circularization, which can form secondary structures and trigger PRR activation [191]. Indeed, IVT circRNAs generated using T4 RNA ligase do not activate the innate immune response. However, it is worth noting that viral IRESs also form secondary structures, potentially contributing to PRR activation. Further investigations are required to ascertain the factors underlying the immunogenicity of in vitro-transcribed circRNA. Such studies should focus on the influence of the primary RNA sequence, secondary structure, size, and the presence of modified nucleosides [192].

By-products of the synthesis of artificial mRNA or circRNA inevitably emerge, including dsDNA, ssDNA, and DNA/RNA heterodimers. Each of these by-products possesses specific receptors that recognize them and activate the innate immune system. For instance, TLR-9, located in the cell endosome, recognizes dsDNA and DNA/RNA hybrids, and cytosolic sensors such as cGAS (cyclic GMP-AMP synthase) and AIM2 (absent in melanoma 2) also detect these molecules, leading to the synthesis of IFN-I and other cytokines [193,194,195,196]. Therefore, it is crucial to purify the IVT mRNA from DNA impurities and DNA/RNA hybrids. DNase-I is effective in eliminating DNA contaminants [176]. However, purifying IVT mRNA from random RNA and dsRNA fragments remains a challenge. A reduction in dsRNA levels can be achieved by decreasing the concentration of Mg2+ ions during in vitro transcription or by incorporating modified nucleotides [173,197]. Moreover, after the synthesis of IVT circRNA, linear RNA molecules have to be removed as well because they are highly immunogenic by-products of the reaction. The treatment of IVT circRNA with RNase R or exonuclease T has been proposed [135,139,198].

Two main approaches are currently employed to purify IVT mRNA and circRNA from dsRNA. The first approach involves reversed-phase high-performance liquid chromatography (HPLC), while the second approach utilizes cellulose [78,199]. Reversed-phase HPLC, however, has several limitations, including poor scalability, toxicity of the eluant (acetonitrile), and considerable losses of IVT mRNA and circRNA [135,190,200]. On the other hand, the cellulose-based method for IVT mRNA purification does not suffer from these drawbacks. It removes up to 90% of dsRNA impurities, though it may not effectively eliminate short RNA fragments [81]. While the use of cellulose to purify IVT circRNA has not been described extensively, it holds potential for such applications. The significance of purifying IVT mRNA from dsRNA impurities is evident from studies demonstrating that purification alone can enhance translated protein levels by 10–1000 times compared to unpurified mRNA [78,166,201]. Similar investigations on IVT CircRNA highlight the critical importance of removing reaction by-products to reduce immunogenicity [131,141,144,188,190,191,192]. 

## 8. Future of mRNA-Based Platform Vaccine

Further development of mRNA vaccines is reduced to the search for a suitable alternative to replace the cap and to find a more suitable variant of the poly A tail. The use of a classical cap or a synthetic analog of the cap seems to be quite financially costly when creating mRNA vaccines. A significant reduction in cost is possible by using IRES, which does not require additional financial costs, instead of the cap. However, at the moment, the mRNA created using IRES is still inferior in terms of protein expression level. Therefore, it seems promising to search for new stronger synthetic analogs of IRES, which will allow us to completely abandon the cap [202].

Another branch of mRNA platform development is self-amplifying mRNA, an improved version of which is represented by trans-amplifying RNA. taRNA will have to be studied in more detail in the future and the main problem related to the safety of its use will have to be solved. The presence of alphavirus enzymes synthesized with taRNA will have to be replaced by enzymes that are safer for humans, and trans-replicon RNA and non-replicating mRNA encoding a replicase will have to be combined into a single mRNA. The last RNA-based vaccine variant in use is circRNA, which has a good chance to become a leader among other RNA platforms in the future. The future development of circRNA is connected with the creation of strong synthetic IRES, which allow us to significantly increase protein production using such RNA [133,203]. A separate direction in the development of this platform is the search for simpler and cheaper ways to synthesize a circular RNA molecule. One of the options is related to the search and use of non-immunogenic ribozymes that allow to create circRNA without additional costs.

## 9. Conclusions

Sixty-two years after its discovery, mRNA has evolved from an object of research into a tool for the prevention and therapy of a number of diseases. Vaccines based on mRNA have become a promising technological platform that is being actively developed. Currently, self-amplifying and trans-amplifying IVT RNA vaccines based on mRNA have been created. In 2015, the discovery of the possibility of translating proteins from circRNA in mammalian cells led to the development of viral vaccines based on them. Each of the four platforms used for vaccine development (mRNA, self-amplifying mRNA, trans-amplifying mRNA, and circRNA) has a number of advantages and disadvantages (Figure 9).

The non-amplifying mRNA platform is the most studied platform for the in vitro synthesis of RNA for its use as a vaccine. It allows for the selection of multiple regulatory elements to increase antigen translation in different cell types and tissues. Various modified bases can also be used to reduce the immunogenicity of the mRNA and prevent immune activation before translation takes place. However, mRNA-based vaccines have a number of disadvantages, mainly the relatively short half-life of the matrix and consequently the small amount of antigen produced. Self-amplifying mRNA is a logical extension of mRNA and is designed to solve its main problems because the mRNA is amplified inside the cell. The amplification of mRNA inside the cell leads to an increase in the initial number of mRNAs inside the cell, and therefore increases the amount of antigen that is translated. Thus, self-amplifying mRNA vaccines can be administered in low doses, have a long half-life inside cells, and result in the production of relatively high amounts of antigen. These unique properties of self-amplifying mRNA are due to the inclusion of a sequence encoding four non-structural proteins (nsP1, nsP2, nsP3, and nsP4) derived from alphaviruses. However, these virus sequences are the main drawbacks of this platform. Since the viral proteins are highly immunogenic and the sequence encoding them is more than 7000 nucleotide pairs, this limits the size of the antigen in such a vaccine. Furthermore, the double-stranded by-products of mRNA amplification are immunogenic. Yet another disadvantage of self-amplifying mRNA is the inability to use modified nitrogenous bases. 

To solve the several problems of the self-amplifying mRNA platform, a trans-amplifying mRNA vaccine platform was created using two kinds of mRNA. The first encodes non-structural proteins of the alphavirus necessary for amplification of the mRNA of interest, which encodes the antigen and contains elements important for its recognition by RNA-dependent RNA polymerase. This platform has all the advantages of self-amplifying mRNA but lacks a number of the disadvantages. The trans-amplifying mRNA platform allows for the use of modified nitrogenous bases, reducing the immunogenicity of the mRNA. But this mRNA platform requires at least two types of mRNA, still contains viral sequences, and is still poorly understood. 

The last type of RNA vaccines is circRNA, which has a unique structure that protects it from degradation by exonucleases and enables it to have a longer half-life, which leads to higher levels of the translated antigen. Furthermore, the circRNA platform does not contain a cap and poly A tail, which pose a number of problems in vaccine development, and allows for the use of modified nitrogenous bases. However, this platform (as well as trans-amplifying mRNA) is still insufficiently studied and the level of synthesized antigen is lower than that in self-replicating mRNA platforms. Nevertheless, at this time, it seems that the circRNA platform is the most promising because, on the one hand, it produces a higher level of antigen than the mRNA platform, and it is also safer than self-replicating mRNAs, which contain viral sequences. The application of any mRNA-based platform is feasible for both therapeutic cancer vaccines and prophylactic vaccines against various viral diseases.

## Figures and Tables

**Figure 1 vaccines-11-01600-f001:**
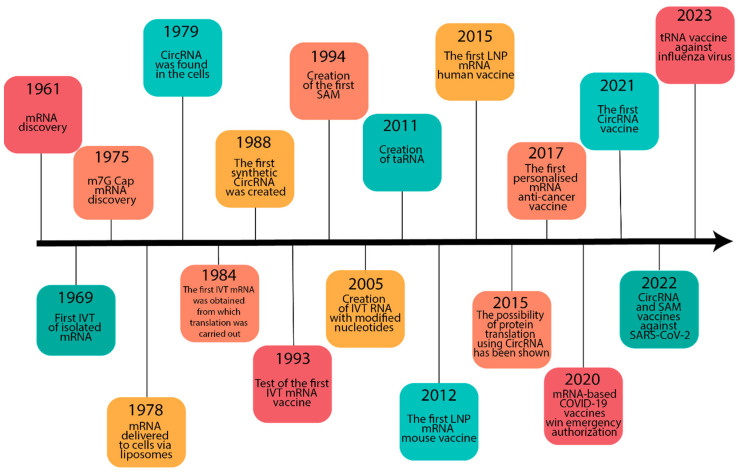
Timeline of some basic discoveries in the development of RNA vaccine technology. Abbreviations: mRNA, messenger RNA; IVT, in vitro-transcribed; circRNA, circular RNA; SAM, self-amplifying mRNA; taRNA, trans-amplifying mRNA; LNP, lipid nanoparticles; COVID-19, coronavirus disease 2019.

**Figure 2 vaccines-11-01600-f002:**
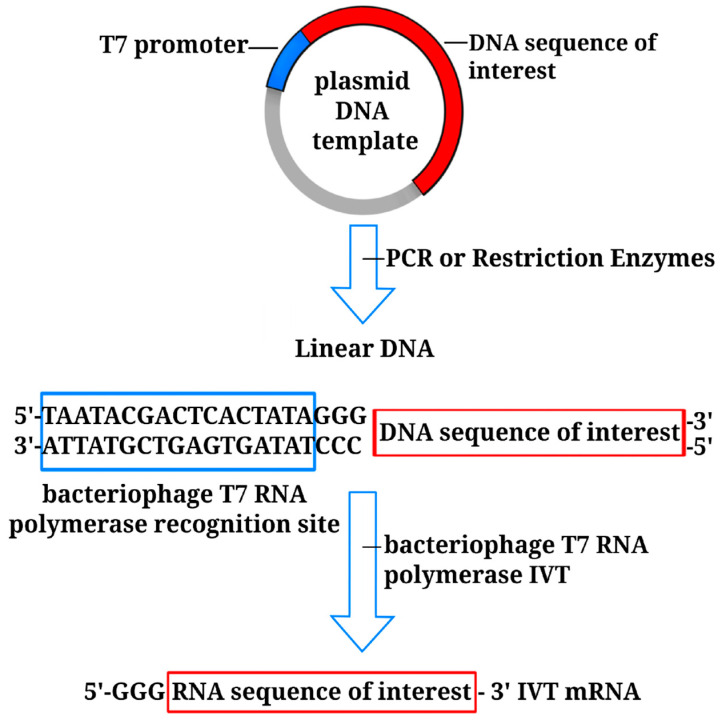
In vitro generation of RNA molecule by a phage T7 RNA polymerase. The blue box represents bacteriophage T7 RNA polymerase recognition site; the red box represents the sequence of interest.

**Figure 3 vaccines-11-01600-f003:**
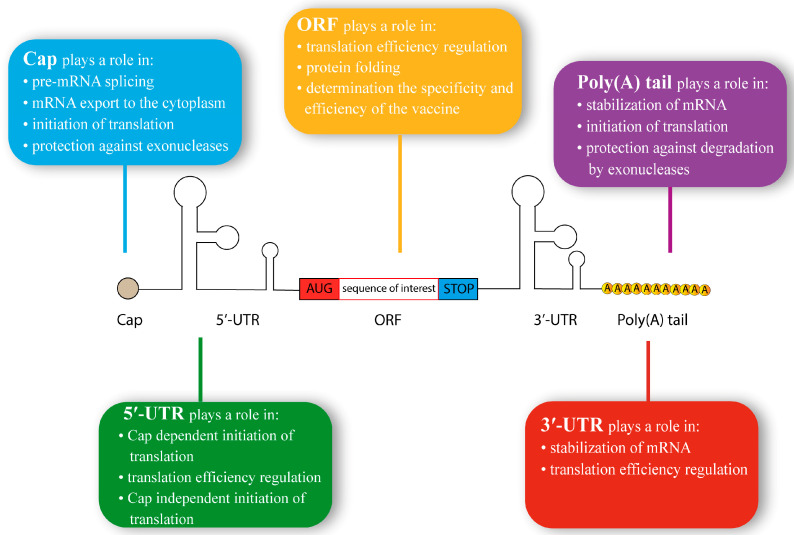
The typical structure of synthetic mRNA. mRNA consists of cap, 5′-untranslated region (5′-UTR), which can include internal ribosome entry sites (IRES), open reading frame (ORF), 3′-untranslated region (3′-UTR), and 3′ poly(A) tail. The colored box characterizes each element in the mRNA structure.

**Figure 4 vaccines-11-01600-f004:**
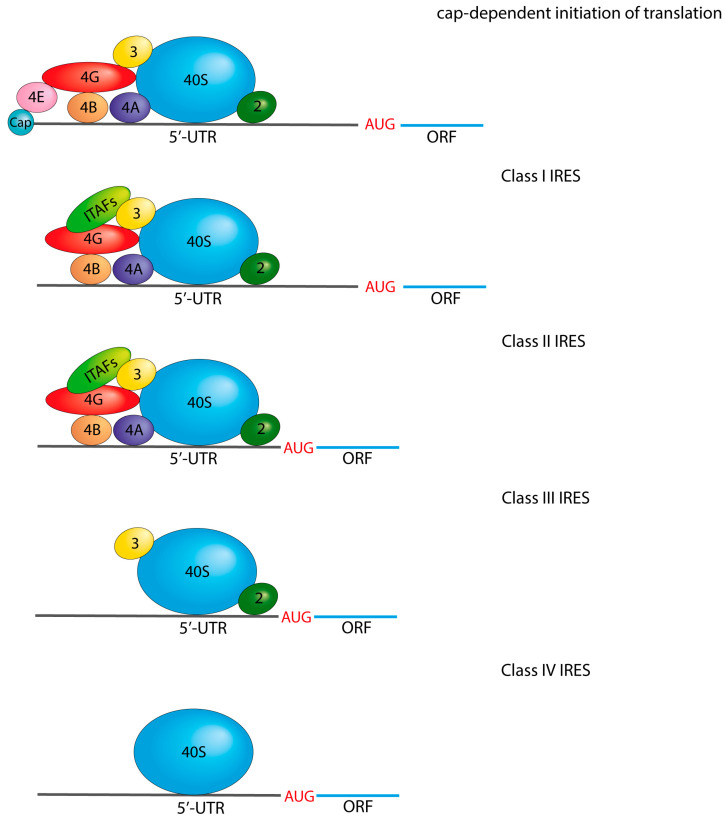
Scheme of cap-dependent and cap-independent initiation of mRNA translation. Cap-dependent translation initiation is the classical way to start the assembly of the eukaryotic initiator complex. The complex includes methionyl initiator tRNA (Met-tRNAi), eukaryotic initiation factor 1 (eIF1), eIF1A, eIF2, eIF3, eIF4A, eIF4B, eIF4G, eIF4E, eIF5, and poly(A)-binding protein (PABP). Class I–IV IRES initiates mRNA translation in a cap-independent way and is typical for viruses. The initiator complex of mRNA translation containing Class I IRES includes all the factors characteristic of the cap-dependent pathway except eIF4E, which recognizes the cap, but adds IRES trans-activating factors (ITAFs). Class II IRES includes all the initiator factors characteristic of Class I IRES, but the initiation itself occurs directly at the AUG codon. IRES class III, similar to IRES class II, attracts translation initiation factors directly to the AUG codon, but only Met-tRNAi, eIF2, and eIF3 perform initiation. Class IV IRES engages the 40S subunit independently without the eIF and AUG codons, promoting translation initiation by structurally mimicking Met-tRNAi.

**Figure 5 vaccines-11-01600-f005:**
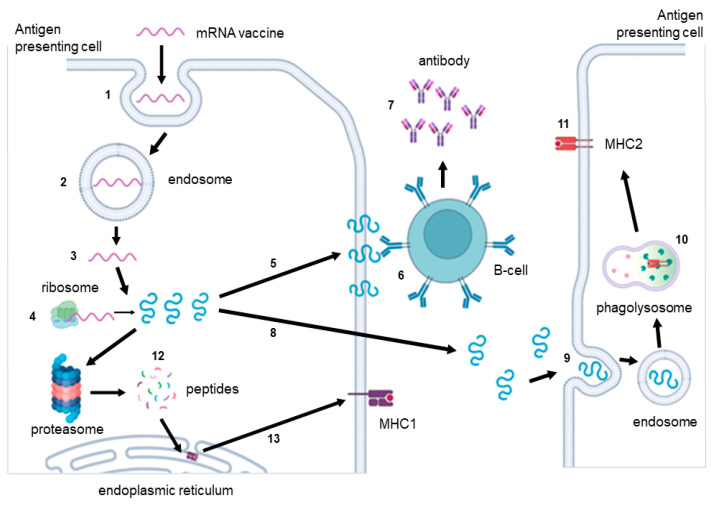
Schematic representation of the mRNA mechanism of action in promoting the adaptive immune response. mRNA is first internalized into the antigen-presenting cell (1) and an endosome containing the mRNA is formed (2). mRNA must exit the endosome into the cytoplasm to avoid degradation in the phagolysosome (3). Once in the cytoplasm, the mRNA is translated by ribosomes (4). Part of the synthesized antigen is transported to the surface of the antigen-presenting cell (5), where it can be recognized by B-lymphocytes (6), which end up forming clones of plasma cells synthesizing antibodies against the antigen (7). Another portion of the synthesized antigen can be secreted into the intercellular space (8), where it can be captured (9) and degraded as part of the phagolysosome (10) of another antigen-presenting cell. After degradation, the antigen is presented as part of the MHC class II (11). The third pathway of the synthesized antigen is proteasomal degradation into peptides within the synthesizing cell (12). The peptides are then presented as part of the MHC class I (13).

**Figure 6 vaccines-11-01600-f006:**
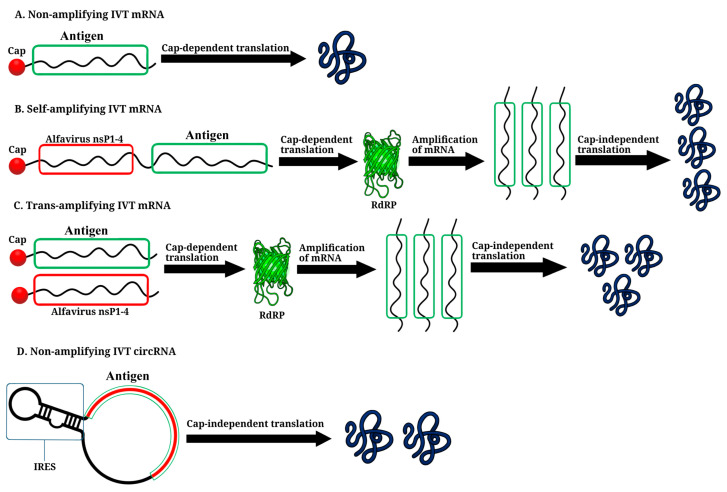
Vaccine design for non-amplifying, self-amplifying, trans-amplifying, and circular in vitro-transcribed RNA (IVT circRNA). Non-amplifying IVT mRNA is the classic RNA vaccine (**A**). A self-replicating RNA vaccine was created on the basis of non-amplifying IVT mRNA by encoding viral RNA-dependent RNA polymerase (RdRp) (**B**). The trans-amplifying IVT mRNA vaccine was the logical extension of self-amplifying IVT mRNA. It includes two types of mRNA: mRNA encoding antigens and mRNA encoding viral replicase (**С**). Circular IVT RNA has a unique closed structure that allows translation initiation only through internal ribosome entry sites (IRES) (**D**). Green box: RNA fragment encoding the antigen conferring the vaccine properties. Red box: RNA fragment encoding a viral RNA-dependent RNA polymerase that can amplify the antigen-encoding site of the RNA to enhance the vaccine action of a specific dose. IRES: internal ribosome entry sites.

**Figure 7 vaccines-11-01600-f007:**
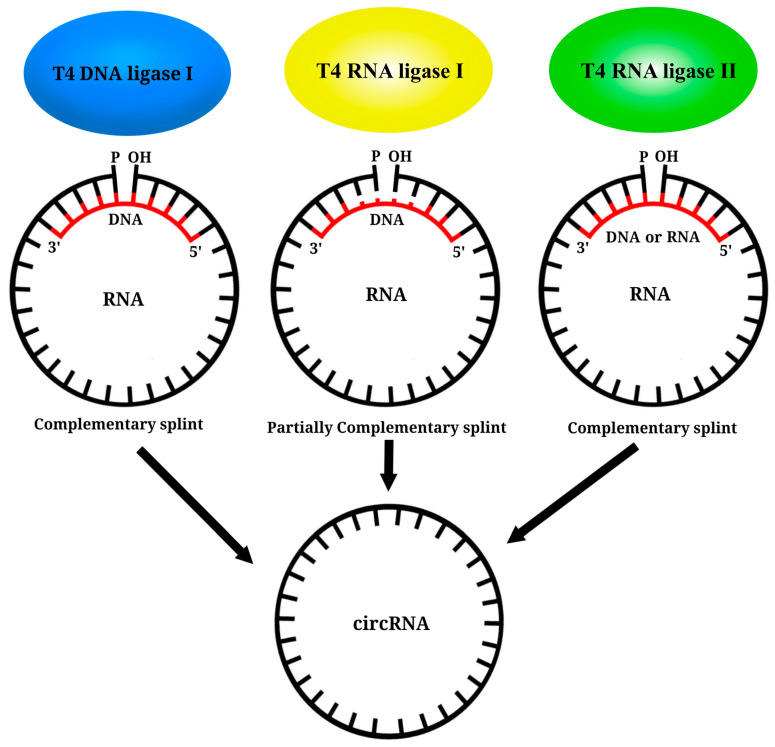
Enzymatic ligation of RNA using T4 ligases. T4 Dnl1 requires the presence of a double-stranded nucleic acid substrate, so a short single-stranded DNA molecule complementary to the circular RNA being synthesized is added to the reaction. T4 Rnl 1 has an affinity for single-stranded RNA molecules. For T4 Rnl 1 to work, the convergence of the joined 3’- and 5’-ends in the RNA molecule is very important, so DNA not fully complementary to the linear RNA molecule is added to the reaction. This DNA splint brings the RNA ends closer together but a few non-complementary terminal nucleotides leave the ends single-stranded. T4 Rnl 2 requires a double-stranded structure to form a circular RNA molecule, so a short RNA or DNA molecule fully complementary to the ends to be joined is added to the reaction.

**Figure 8 vaccines-11-01600-f008:**
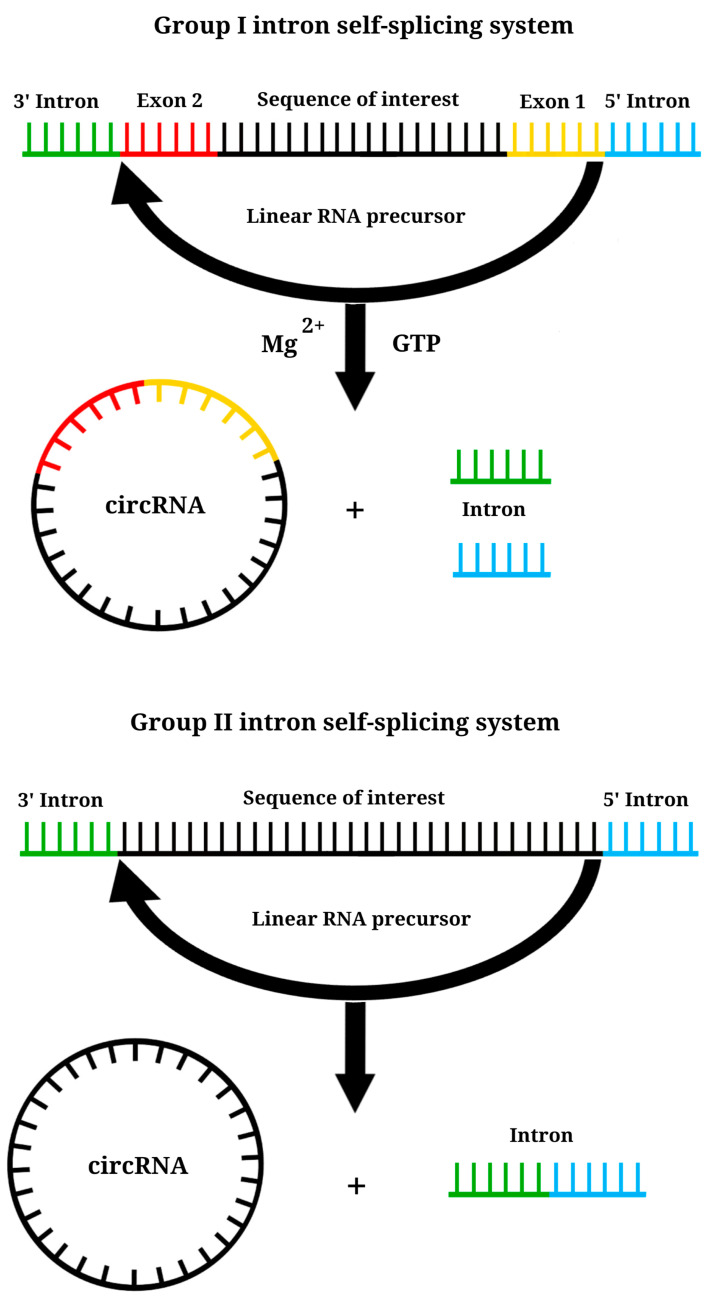
Ribozyme methods for circRNA synthesis. The group I intron self-splicing system comprises RNA fragments of exons and introns of the foreign organism flanking the antigen sequence. The group I intron self-splicing system requires guanosine triphosphate (GTP) and Mg2+ in the reaction as co-factors for the formation of circRNA from a linear RNA molecule. The principle of the group II intron self-splicing method resembles the principle of the group I intron self-splicing method, but it can use foreign introns without exons. However, the self-splicing of group II introns is known to form 2’, 5’-phosphodiester bonds at the ligation site.

**Figure 9 vaccines-11-01600-f009:**
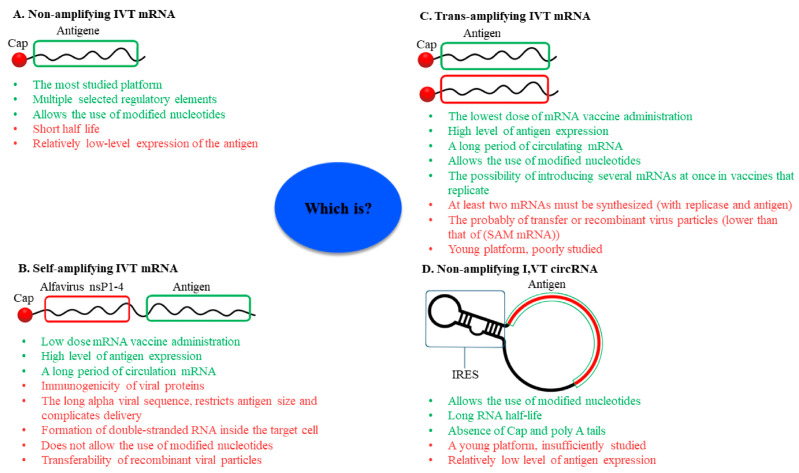
Advantages and disadvantages of different RNA vaccine platforms. Non-amplifying IVT mRNA platform (**A**), self-amplifying IVT mRNA (**B**), trans-amplifying IVT mRNA (**C**), IVT circRNA (**D**). Green text represents advantages and red text indicates disadvantages of the RNA platform. IVT: in vitro-transcribed, IRES: internal ribosome entry sites.

**Table 1 vaccines-11-01600-t001:** Basic cap structures used in IVT mRNA modification [24,26,30,34,35,36,37,38].

Type of Cap	Structure	Capping Enzymes	Capping Stage
Cap 0	m7G-ppp-N-p	Vaccinia Capping Enzyme (VCE)	Separate stage
Cap 1	m7G-ppp-N2’-Me-Op	Vaccinia Capping Enzyme (VCE) and mRNA cap 2′O-methyltransferase (MTase)	Two separate stages
Anti-Reverse Cap Analog (ARCA)	3′-O-Me-m7G5′-ppp-5′G	RNA polymerases of bacteriophages T3, T7 or SP6	Сo-transcriptionally
CleanCap^®^ Reagent AG 3′OMe	m7G3′-O-Me-ppp-A2′-O-Me-p-G	RNA polymerases of bacteriophages T3, T7 or SP6	Сo-transcriptionally
CleanCap^®^ Reagent AG	m7G-ppp-A2′-O-Me-p-G	RNA polymerases of bacteriophages T3, T7 or SP6	Сo-transcriptionally
CleanCap^®^ Reagent AU	m7G-ppp-A2′-O-Me-p-U	RNA polymerases of bacteriophages T3, T7 or SP6	Сo-transcriptionally

Abbreviations: p is a phosphate, N is any nucleotide, m7G is 7 methyl guanosine, O is oxygen, and Me is a methyl group.

**Table 2 vaccines-11-01600-t002:** Basic pattern recognition receptors (PRRs) that recognize in vitro transcription nuclear acids [78,166,167,168,169].

Nucleic Acid	PRR	Effects
ssRNA	TLR7	Inhibition of protein translation; Synthesis of pro-inflammatory cytokines
	TLR8	Synthesis of pro-inflammatory cytokines
	NOD2	Synthesis of pro-inflammatory cytokines
dsRNA	TLR3	Inhibition of protein translation; Synthesis of pro-inflammatory cytokines
	RIG-I	Inhibition of protein translation; Synthesis of pro-inflammatory cytokines
	MDA-5	Inhibition of protein translation; Synthesis of pro-inflammatory cytokines
	NLRP3	Synthesis of pro-inflammatory cytokines
	PKR	Stop of translation
	OAS	RNA degradation
CircRNA	RIG-I	Inhibition of protein translation; Synthesis of pro-inflammatory cytokines
	PKR	Stop of translation
dsDNA, RNA/DNA	TLR9	Synthesis of pro-inflammatory cytokines
	cGAS	Synthesis of pro-inflammatory cytokines
	AIM2	Synthesis of pro-inflammatory cytokines

Abbreviations: TLR, Toll-like receptor; NOD, nucleotide oligomerization domain; RIG, retinoic acid-inducible gene-I; MDA, melanoma differentiation-associated gene 5; NLR, nucleotide oligomerization domain (NOD)-like receptor; PKR, RNA-dependent protein kinase; OAS, oligoadenylate synthetase; cGAS, cyclic GMP-AMP synthase; AIM2, absent in melanoma 2.

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
