# Peer review of "In Vitro Transcribed RNA-Based Platform Vaccines: Past, Present, and Future"

_vaccines, 2023, doi:10.3390/vaccines11101600_

Round 1

Reviewer 1 Report

There are issues with the papers cited by the authors use to corroborate the text. For

example in the sentences in lines 79-83, the authors say that CircRNA was discovered

in 1979, but cite the paper by Bloom et al (2021), which does not mention anything

about CircRNA’s discovery in 1979. In fact Sanger et al in 1976 first described about

circRNA, and Hsu and Coca-parados in 1979 reported the electron microscopic

evidence of circRNA. The authors’ should cite these papers instead of the paper by

Bloom et al. Next, the authors write about the first synthesis of circRNA in 1988 by

Melton and Krieg but cite a paper by Rio D.C, 2013. The authors should have cited

Melton and Krieg (1987). See https://doi.org/10.1073/pnas.73.11.3852,

https://doi.org/10.1038/280339a0, and https://doi.org/10.1016/0076-6879(87)55027-

3. The authors are suggested to revise the manuscript and add relevant citations.

ï‚· Again in lines 378-380 the authors mention “For example, the mitochondrially

encoded 12S rRNA (mtRNR1) has shown high protein levels when used alone or in

combination with an amino-terminal enhancer of split (AES) gene fragment” and cite

Siepel et al, 2005. But the cited paper does not provide this information.

ï‚· Line 105-106 “In 2011, a significant advancement in mRNA technology known as

SAM was achieved with the creation of the first trans-amplifying mRNA (taRNA)”.

Here, citation is missing.

ï‚· Line 172, 447 Italicize the scientific names.

ï‚· Add references to Table 1 and Table 2.

ï‚· Line 345-346: “This computational approach can predict 3an entire proteins”. What

are the authors implying?

ï‚· Figure 6 and Figure 9: The authors have misspelled antigen as “antigene”. Figure 8:

Correct spelling of intron.

ï‚· Include a section to discuss future prospective about various aspects of self-

amplifying mRNA vaccine, trans-amplifying mRNA vaccines and circRNA vaccines.

It is fine. 

Reviewer 2 Report

I believe this review article can be accepted with minor revisions. Here are some positive aspects and minor comments for improvement:

Positive Aspects:

The abstract provides a clear and concise overview of the review article's content, including the historical background of mRNA vaccines and the emergence of self-amplifying mRNA, trans-amplifying mRNA, and circular RNA platforms.

The keywords are relevant and well-defined, helping readers understand the focus of the article.

The introduction provides context by comparing traditional and modern vaccines, highlighting the advantages of mRNA vaccines, and setting the stage for the discussion of different mRNA platforms.

The historical background section gives a brief but informative overview of the development of mRNA as a vaccine technology, providing readers with essential context.

Minor Comments for Improvement:

In the introduction, consider citing recent examples of mRNA vaccines, such as COVID-19 vaccines, to emphasize the timeliness and relevance of the topic.

In the historical background section, you could expand on the challenges and limitations faced by mRNA vaccines, which would provide a more comprehensive historical perspective.

The abstract does not specify the length of the review article. Including this information could help readers understand the depth and scope of the review.

Consider mentioning the potential applications or diseases that could benefit from each of the mRNA platforms (mRNA, self-amplifying mRNA, trans-amplifying mRNA, circular RNA) to make the article more practical for researchers and clinicians.

In the historical background section, it might be beneficial to briefly mention the development of mRNA vaccines against COVID-19, as this recent milestone has significantly impacted the field.

Reviewer 3 Report

The manuscript is an extensive review on the various mRNA-based vaccine platforms developed.  The authors have described in details relevant background, underlying mechanism of action, different mRNA vaccine platforms, immunogenicity, and future perspectives.  The review is well written, detailed, and informative. The figures are very useful as they make the concepts easy to understand.  Overall, the manuscript is in good shape.

Author Response

Thank you for your attention and appreciation for our manuscript. Thank you for providing an opportunity to publish it.

Reviewer 4 Report

The Nobel Prize in Physiology or Medicine 2023 was awarded jointly to Katalin Karikó and Drew Weissman "for their discoveries concerning nucleoside base modifications that enabled the development of effective mRNA vaccines against COVID-19". The speed with which mRNA vaccines can be developed also paves the way for the development of drugs against other infectious diseases. In the future, this technology could be used for drug delivery and cancer treatment. In the review article, the authors discuss the characteristics of several mRNA platforms that are potentially applicable for the development of mRNA vaccines against cancer or viral diseases.

Minor concern: Fig.3, violet lable, last line needs to be continued. Presumably, "protects against degradation by exonucleases"
